# OmniPrint:
# A Configurable Printed Character Synthesizer

**Haozhe Sun**[*], **Wei-Wei Tu**[#+], **Isabelle Guyon**[*+]

[*] LISN (CNRS/INRIA) Université Paris-Saclay, France
[#] 4Paradigm Inc, Beijing, China
[+] ChaLearn, California, USA
`omniprint@chalearn.org`

## Abstract

We introduce OmniPrint, a synthetic data generator of isolated printed characters, geared toward machine learning research. It draws inspiration from famous datasets such as MNIST, SVHN and Omniglot, but offers the capability of generating a wide variety of printed characters from various languages, fonts and styles, with customized distortions. We include 935 fonts from 27 scripts and many types of distortions. As a proof of concept, we show various use cases, including an example of meta-learning dataset designed for the upcoming MetaDL NeurIPS 2021 competition. OmniPrint is available at https://github.com/SunHaozhe/OmniPrint.

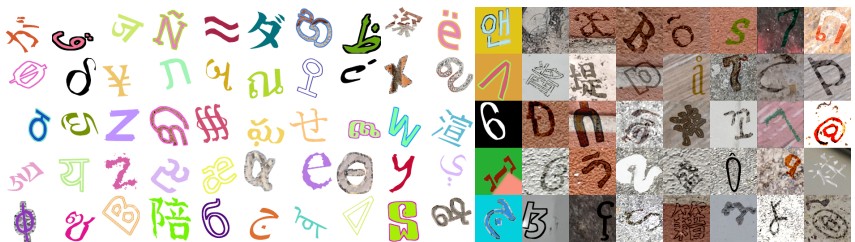

Figure 1: **Examples of characters generated by OmniPrint.**

## 1 Introduction and motivation

Benchmarks and shared datasets have been fostering progress in Machine learning (ML) [53, 37, 68, 28]. One of the most popular benchmarks is MNIST [39], which is used all over the world in tutorials, textbooks, and classes. Many variants of MNIST have been created [10, 8, 47, 59, 18, 71], including one created recently, which inspired us: Omniglot [38]. This dataset includes characters from many different scripts. Among machine learning techniques using such benchmark datasets, Deep Learning techniques are known to be very data hungry. Thus, while there is an increasing number of available datasets, there is a need for larger ones. But, collecting and labeling data is time consuming and expensive, and systematically varying environment conditions is difficult and necessarily limited. Therefore, resorting to artificially generated data is useful to drive fundamental research in ML. This motivated us to create OmniPrint, as an extension to Omniglot, geared to the generation of an unlimited amount of printed characters.

Of all ML problems, we direct our attention to classification and regression problems in which a vector **y** (discrete or continuous labels) must be predicted from a real-valued input vector **x** of observations (in the case of OmniPrint, an image of a printed character). Additionally, data are plagued by nuisance variables **z**, another vector of discrete or continuous labels, called *metadata* or *covariates*. In the problem at hand, **z** may include various character distortions, such as shear, rotation, line width variations, and changes in background. Using capital letters for random variable

35th Conference on Neural Information Processing Systems (NeurIPS 2021) Track on Datasets and Benchmarks.

and lowercase for their associated realizations, a data generating process supported by OmniPrint consists in three steps:

$$\mathbf{z} \quad \sim \quad \mathbb{P}(\mathbf{Z}) \tag{1}$$
$$\mathbf{y} \quad \sim \quad \mathbb{P}(\mathbf{Y}|\mathbf{Z}) \tag{2}$$
$$\mathbf{x} \quad \sim \quad \mathbb{P}(\mathbf{X}|\mathbf{Z}, \mathbf{Y}) \tag{3}$$

Oftentimes, $\mathbf{Z}$ and $\mathbf{Y}$ are independent, so $\mathbb{P}(\mathbf{Y}|\mathbf{Z}) = \mathbb{P}(\mathbf{Y})$. This type of data generating process is encountered in many domains such as image, video, sound, and text applications (in which objects or concepts are target values $\mathbf{y}$ to be predicted from percepts $\mathbf{x}$); medical diagnoses of genetic disease (for which $\mathbf{x}$ is a phenotype and $\mathbf{y}$ a genotype); analytical chemistry (for which $\mathbf{x}$ may be chromatograms, mass spectra, or other instrument measurements, and $\mathbf{y}$ compounds to be identified), etc. Thus, we anticipate that progress made using OmniPrint to benchmark machine learning systems should also foster progress in these other domains.

Casting the problem in such a generic way should allow researchers to target a variety of ML research topics. Indeed, character images provide excellent benchmarks for machine learning problems because of their relative simplicity, their visual nature, while opening the door to high-impact real-life applications. However, our survey of available resources (Section 2) revealed that no publicly available data synthesizer fully suits our purposes: generating realistic quality images $\mathbf{x}$ of small sizes (to allow fast experimentation) for a wide variety of characters $\mathbf{y}$ (to study extreme number of classes), and wide variety of conditions parameterized by $\mathbf{z}$ (to study invariance to realistic distortions). A conjunction of technical features is required to meet our specifications: pre-rasterization manipulation of anchor points; post-rasterization distortions; natural background and seamless blending; foreground filling; anti-aliasing rendering; importing new fonts and styles.

Modern fonts (*e.g.,* TrueType or OpenType) are made of straight line segments and quadratic Bézier curves, connecting anchor points. Thus it is easy to modify characters by moving anchor points. This allows users to perform vectors-space pre-rasterization geometric transforms (rotation, shear, etc.) as well as distortions (*e.g.,* modifying the length of ascenders of descenders), without incurring aberrations due to aliasing, when transformations are done in pixel space (post-rasterization). The closest software that we found fitting our needs is "Text Recognition Data Generator" [2] (under MIT license), which we used as a basis to develop OmniPrint. While keeping the original software architecture, we substituted individual components to fit our needs. Our contributions include: (1) Implementing many **new transformations and styles**, *e.g.,* elastic distortions, natural background, foreground filling, etc.; (2) Manually selecting characters from the Unicode standard to form alphabets from **more than 20 languages around the world**, further grouped into partitions, to facilitate creating meta-learning tasks; (3) Carefully **identifying fonts**, which suit these characters; (4) Replacing character rendering by a low-level FreeType font rasterization engine [62], which enables **direct manipulation of anchor points**; (5) Adding **anti-aliasing rendering**; (6) Implementing and optimizing utility code to facilitate **dataset formatting**; (7) Providing a meta-learning **use case** with a sample dataset. To our knowledge, OmniPrint is the first text image synthesizer geared toward ML research, supporting pre-rasterization transforms. This allows Omniprint to imitate handwritten characters, to some degree.

## 2 Related work

While our focus is on generating isolated characters for ML research, related work is found in OCR research and briefly reviewed here. OCR problems include **recognition of text from scanned documents** and **recognition of characters "in the wild"** from pictures of natural scenes:

**- OCR from scanned documents** is a well developed field. There are many systems performing very well on this problem [49, 32, 5]. Fueling this research, many authors have addressed the problem of generating artificial or semi-artificial degraded text since the early 90's [33]. More recently, Kieu *et al.* [36] simulate the degradation of aging document and the printing/writing process, such as dark specks near characters or ink discontinuities, and Kieu *et al.* [34] extend this work by facilitating the parameterization. Liang *et al.* [40] generalize the perspective distortion model of Kanungo *et al.* [33] by modeling thick and bound documents as developable surfaces. Kieu *et al.* [35] present a 3D model for reproducing geometric distortions such as folds, torns or convexo-concaves of the paper sheet. Besides printed text, handwritten text synthesis has also been investigated, *e.g.,* [20].

**- Text in the wild, or scene text**, refer to text captured in natural environments, such as sign boards, street signs, etc. yielding larger variability in size, layout, background, and imaging conditions. Contrary to OCR in scanned documents, scene text analysis remains challenging. Furthermore, the size of existing real scene text datasets is still small compared to the demand of deep learning models. Thus, synthetic data generation is an active field of research [5]. Early works did not use deep learning for image synthesis. They relied on font manipulation and traditional image processing techniques, synthetic images are typically rendered through several steps including font rendering, coloring, perspective transformation, background blending, etc. [13, 66, 32]. In recent years, text image synthesis involving deep learning has generated impressive and photo-realistic text in complex natural environments [21, 49, 23, 4, 76, 46, 78, 74, 77, 70, 73]. We surveyed the Internet for open-source text generation engines. The most popular ones include SynthText [23], UnrealText [46], TextRecognitionDataGenerator [2], Text Generator [24], Chinese OCR synthetic data [67], Text Renderer [7] and the Style-Text package of PaddleOCR [70, 14].

As a result of these works, training solely on synthetic data has become a widely accepted practice. Synthetic data alone is sufficient to train state-of-the-art models for the scene text recognition task (tested on real data) [1, 48, 46]. However, despite good performance on existing real evaluation datasets, some limitations have been identified, including failures on longer characters, smaller sizes and unseen font styles [5], and focus on Latin (especially English) or Chinese text [55, 42]. Recently, more attention has been given to these problems [30, 5, 72, 48, 54, 4]. OmniPrint is helpful to generate small-sized quality text image data, covering extreme distortions in a wide variety of scripts, while giving full control over the choice of distortion parameters, although no special effort has been made, so far, to make such distortions fully realistic to immitate characters in the wild.

## 3 The OmniPrint data synthesizer

### 3.1 Overview

OmniPrint is based on the open source software TextRecognitionDataGenerator [2]. While the overall architecture was kept, the software was adapted to meet our required specifications (Table 1 and Figure 2). To obtain a large number of classes (**Y** labels), we **manually collected and filtered characters** from the Unicode standard in order to form alphabets covering more than 20 languages around the world, these alphabets are further divided into partitions *e.g.,* characters from the Oriya script are partitioned into Oriya consonants, Oriya independent vowels and Oriya digits. Nuisance parameters **Z** were decomposed into **Font, Style, Background, and Noise**. To obtain a variety of fonts, we provided an **automatic font collection module**, this module filters problematic fonts and provides fonts' metadata. To obtain a variety of "styles", we substituted the low-level text rendering process by the **FreeType rasterization engine** [62]. This enables **vector-based pre-rasterization transformations**, which are difficult to do with pixel images, such as natural random elastic transformation, stroke width variation and modifications of character proportion (*e.g.,* length of ascenders and descenders). We enriched **background generation** with seamless background blending [52, 23, 22]. We proposed a framework for inserting **custom post-rasterization transformations** (*e.g.,* perspective transformations, blurring, contrast and brightness variation). Lastly, we implemented **utility** code including dataset formatters, which convert data to AutoML format [44] or AutoDL File format [43], to facilitate the use of such datasets in challenges and benchmarks, and a data loader which generates episodes for meta-learning application.

### 3.2 Technical aspects of the design

OmniPrint has been designed to be **extensible**, such that users can easily add new alphabets, new fonts and new transformations into the generation pipeline, see Appendix C, Appendix D and Appendix E. Briefly, here are some highlights of the pipeline of Figure 2:

1. **Parameter configuration file:** We support both TrueType or OpenType font files. Style parameters include rotation angle, shear, stroke width, foreground, text outline and other transformation-specific parameters.

2. **FreeType vector representation:** The chosen text, font and style parameters are used as the input to the FreeType rasterization engine [62].

Table 1: **Comparison of TextRecognitionDataGenerator [2] and OmniPrint.**

| | TRDG [2] | OmniPrint [ours] |
|---|---|---|
| Number of characters | 0 | $12,729$ |
| Number of words | $\simeq 11,077,866$ | 0 |
| Number of fonts | 105 | 935 + automatic font collection |
| Pre-rasterization transforms | 0 | 7 (including elastic distortions) |
| Post-rasterization transforms | 6 | 15 (+ anti-aliasing rendering) |
| Foreground | black | color, outline, natural texture |
| Background | speckle noise, quasicrystal, white, natural image | same plus seamless blending [52, 23, 22] of foreground on background |
| Code organization | Transformations hard-coded | Parameter configuration file, Module plug-ins |
| Dataset formatting | None | Metadata recording, standard format support [43, 44], multi-alphabet support, episode generation |

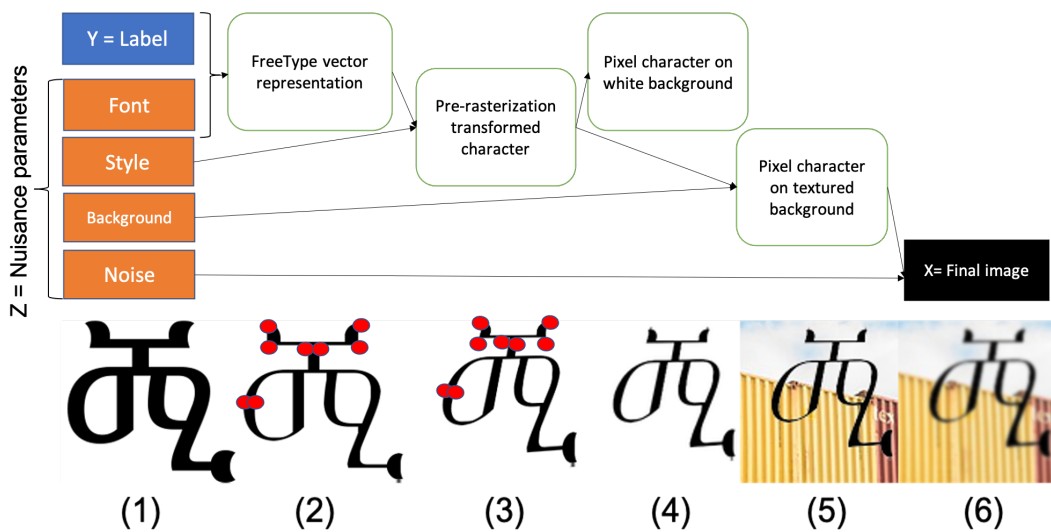

Figure 2: **Basic character image generative process.** The generative process produces images $\mathbf{X}$ as a function of $\mathbf{Y}$ (label or character class) and $\mathbf{Z}$ (nuisance parameter). Only a subset of anchor point (red dots) are shown in steps (2) and (3). A subset of nuisance parameters are chosen for illustration.

3. **Pre-rasterization transformed character:** FreeType also performs all the pre-rasterization (vector-based) transformations, which include linear transforms, stroke width variation, random elastic transformation and variation of character proportion. The RGB bitmaps output by FreeType are called the foreground layer.

4. **Pixel character on white background:** Post-rasterization transformations are applied to the foreground layer. The foreground layer is kept at high resolution at this stage to avoid introducing artifacts. The RGB image is then resized to the desired size with anti-aliasing techniques. The resizing pipeline consists of three steps: (1) applying Gaussian filter to smooth the image; (2) reducing the image by integer times; (3) resizing the image using Lanczos resampling. The second step of the resizing pipeline is an optimization technique proposed by the PIL library [9].

5. **Pixel character on textured background:** The resized foreground layer is then pasted onto the background at the desired position.

6. **Final image:** Some other post-rasterization transformations may be applied after adding the background *e.g.,* Gaussian blur of the whole image. Before outputting the synthesized text image, the image mode can be changed if needed (*e.g.,* changed to grayscale or binary images).

Labels $\mathbf{Y}$ (isolated characters of text) and nuisance parameters $\mathbf{Z}$ (font, style, background, etc.) are output together with image $\mathbf{X}$. $\mathbf{Z}$ serve as "metadata" to help diagnose learning algorithms. The role of $\mathbf{Y}$ and (a subset of) $\mathbf{Z}$ may be exchanged to create a variety of classification problems (*e.g.,* classifying alphabets or fonts), or regression problems (*e.g.,* predicting rotation angles or shear).

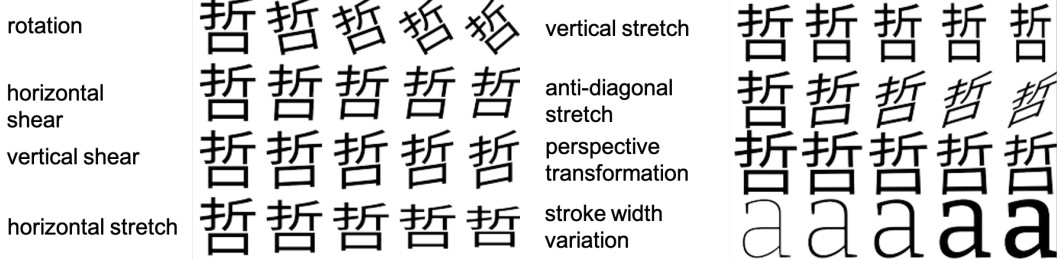

Figure 3: **Some implemented transformations.**

## 3.3 Coverage

We rely on the Unicode 9.0.0 standard [11], which consists of a total of 128172 characters from more than 135 scripts, to identify characters by "code point". A code point is an integer, which represents a single character or part of a character; some code points can be chained to represent a single character *e.g.,* the small Latin letter o with circumflex ô can be either represented by a single code point 244 or a sequence of code points (111, 770), where 111 corresponds to the small Latin letter o, 770 means combining circumflex accent. In this work, we use NFC normalized code points [69] to ensure that each character is uniquely identified.

We have included 27 scripts: Arabic, Armenian, Balinese, Bengali, Chinese, Devanagari, Ethiopic, Georgian, Greek, Gujarati, Hebrew, Hiragana, Katakana, Khmer, Korean, Lao, Latin, Mongolian, Myanmar, N'Ko, Oriya, Russian, Sinhala, Tamil, Telugu, Thai, Tibetan. For each of these scripts, we manually selected characters. Besides skipping unassigned code points, control characters, incomplete characters, we also filtered Diacritics, tone marks, repetition marks, vocalic modification, subjoined consonants, cantillation marks, etc. For Chinese and Korean characters, we included the most commonly used ones. Details on the selection criteria are given in Appendix F.

The fonts in the first release of OmniPrint have been collected from the Internet. In total, we have selected 12729 characters from 27 scripts (and some special symbols) and 935 fonts. The details of alphabets, fonts can be found in Appendix C and Appendix F.

## 3.4 Transformations

Examples of transformations that we implemented are shown in Figure 3. We are interested in all "label-preserving" transformations on text images as well as their compositions. A transformation is said to be label-preserving if applying it does not alter the semantic meaning of the text image, as interpreted by a human reader. The pre- and post-rasterization transformations that we implemented are detailed in Appendix D and Appendix E.

They are classified as **geometric transformations** (number 1-4: each class is a subset of the next class), **local transformations**, and **noises**:

1. **Isometries: rotation, translation.** Isometries are bijective maps between two metric spaces that preserve distances, they preserve lengths, angles and areas. In our case, rotation has to be constrained to a certain range in order to be label-preserving, the exact range of rotation may vary in function of scripts. Reflection is not desired because it is usually not label-preserving for text images. For human readers, a reflected character may not be acceptable or may even be recognized as another character.

2. **Similarities: uniform scaling.** Similarities preserve angles and ratios between distances. Uniform scaling includes enlarging or reducing.

3. **Affine transformations: shear, stretch.** Affine transformations preserve parallelism. Shear (also known as skew, slant, oblique) can be done either along horizontal axis or vertical axis. Stretch is usually done along the four axes: horizontal axis, vertical axis, main diagonal and anti-diagonal axis. Stretch can be seen as non-uniform scaling. Stretch along horizontal or vertical axis is also referred to as parallel hyperbolic transformation, stretch along main diagonal or anti-diagonal axis is also referred to as diagonal hyperbolic transformation [56].

Table 2: **Comparison of Omniglot and OmniPrint.**

|  | Omniglot [38] | OmniPrint [ours] |
|---|---|---|
| Total number of unique characters (classes) | 1623 | Unlimited (1409 in our example) |
| Total number of examples | 1623×20 | Unlimited (1409×20 in our example) |
| Number of possible alphabets (super-classes) | 50 | Unlimited (54 in our example) |
| Scalability of characters and super-classes | No | Yes |
| Diverse transformations | No | Yes |
| Natural background | No | Yes (OmniPrint-meta5 in our exemple) |
| Possibility of increasing image resolution | No | Yes |
| Performance of Prototypical Networks 5-way-1-shot [58] | 98.8% | 61.5%-97.6% |
| Performance of MAML 5-way-1-shot [15] | 98.7% | 63.4%-95.0% |

Table 3: **OmniPrint-meta[1-5] datasets** of progressive difficulty. Elastic means random elastic transformations. Fonts are sampled from all the fonts available for each character set. Transformations include random rotation (within -30 and 30 degrees), horizontal shear and perspective transformation.

| X | Elastic | # Fonts | Transformations | Foreground | Background |
|---|---|---|---|---|---|
| 1 | **Yes** | 1 | No | Black | White |
| 2 | **Yes** | **Sampled** | No | Black | White |
| 3 | **Yes** | **Sampled** | **Yes** | Black | White |
| 4 | **Yes** | **Sampled** | **Yes** | **Colored** | Colored |
| 5 | **Yes** | **Sampled** | **Yes** | **Colored** | **Textured** |

4. **Perspective transformations.** Perspective transformations (also known as homographies or projective transformations) preserve collinearity. This transformation can be used to imitate camera viewpoint *i.e.,* 2D projection of 3D world.

5. **Local transformations**: Independent random vibration of the anchor points. Variation of the stroke width *e.g.,* thinning or thickening of the strokes. Variation of character proportion *e.g.,* length of ascenders and descenders.

6. **Noises** related to imaging conditions *e.g.,* Gaussian blur, contrast or brightness variation.

# 4 Use cases

## 4.1 Few-shot learning

We present a first use case motivated by the organization of the NeurIPS 2021 meta-learning challenge (MetaDL). We use OmniPrint to generate several few-shot learning tasks. Similar datasets are used in the challenge.

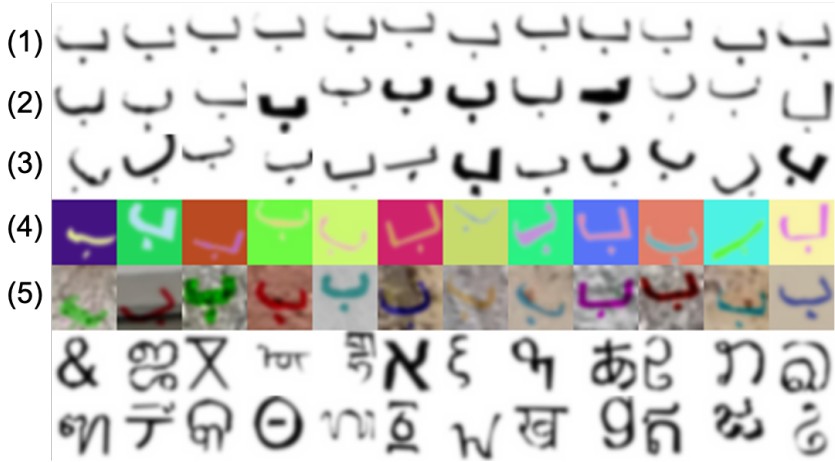

Figure 4: **OmniPrint-meta[1-5] sample data:** Top: The same character of increasing difficulty. Bottom: Examples of characters showing the diversity of the 54 super-classes.

Few-shot learning is a ML problem in which new classification problems must be learned "quickly", from just a few training examples per class (shots). This problem is particularly important in domains in which few labeled training examples are available, and/or in which training on new classes must be done quickly episodically (for example if an agent is constantly exposed to new environments). We chose OmniPrint as one application domain of interest for few-shot learning. Indeed, alphabets from many countries are seldom studied and have no dedicated OCR products available. A few-shot learning recognizer could remedy this situation by allowing users to add new alphabets with *e.g.,* a single example of each character of a given font, yet generalize to other fonts or styles.

Recently, interest in few-shot learning has been revived (*e.g.,* [15, 58, 27]) and a novel setting proposed. The overall problem is divided into many sub-problems, called episodes. Data are split for each episode into a pair $\{support\ set,\ query\ set\}$. The support set plays the role of a training set and the query set that of a test set. In the simplified research setting, each episode is supposed to have the same number $N$ of classes (characters), also called **"ways"**. For each episode, learning machines receive $K$ training examples per class, also called **"shots"**, in the "support set"; and a number of test examples from the same classes in the "query set". This yields a $N$-**way**-$K$-**shot problem**. To perform meta-learning, data are divided between a **meta-train set** and a **meta-test set**. In the meta-train set, the support and query set labels are visible to learning machines; in contrast, in the meta-test set, only support set labels are visible to learning machines; query set labels are concealed and only used to evaluate performance. In some few-shot learning datasets, classes have hierarchical structures [61, 38, 53] *i.e.,* classes sharing certain semantics are grouped into super-classes (which can be thought of as alphabets or partitions of alphabets in the case of OmniPrint). In such cases, episodes can coincide with super-classes, and may have a variable number of "ways".

Using OmniPrint to benchmark few-shot learning methods was inspired by Omniglot [38], a popular benchmark in this field. A typical way of using Omniglot is to pool all characters from different alphabets and sample subsets of $N$ characters to create episodes (*e.g.,* $N = 5$ and $K = 1$ results in a 5-way-1-shot problem). While Omniglot has fostered progress, it can hardly push further the state-of-the-art since recent methods, *e.g.,* MAML [15] and Prototypical Networks [58] achieve a classification accuracy of $98.7\%$ and $98.8\%$ respectively in the 5-way-1-shot setting. Furthermore, Omniglot was not intended to be a realistic dataset: the characters were drawn online and do not look natural. In contrast OmniPrint provides realistic data with a variability encountered in the real world, allowing us to **create more challenging tasks**. We compare Omniglot and OmniPrint for few-shot learning benchmarking in Table 2.

As a proof of concept, we created 5 datasets called OmniPrint-meta[1-5] of progressive difficulty, from which few-shot learning tasks can be carved (Table 3 and Figure 4). These 5 datasets imitate the setting of Omniglot, for easier comparison and to facilitate replacing it as a benchmark. The OmniPrint-meta[1-5] datasets share the same set of 1409 characters (classes) from 54 super-classes, with 20 examples each, but they **differ in transformations and styles**. Transformations and distortions are cumulated from dataset to dataset, each one including additional transformations to make characters harder to recognize. We synthesized $32 \times 32$ RGB images of isolated characters. The datasheet for dataset [19] for the OmniPrint-meta[1-5] datasets is shown in Appendix A.

We performed a few learning experiments with classical few-shot-learning baseline methods: Prototypical Networks [58] and MAML [15] (Table 4). The naive baseline trains a neural network from scratch for each meta-test episode with 20 gradient steps. We split the data into 900 characters for meta-train, 149 characters for meta-validation, 360 characters for meta-test. The model having the highest accuracy on meta-validation episodes during training is selected to be tested on meta-test episodes. Performance is evaluated with the average classification accuracy over 1000 randomly generated meta-test episodes. The reported accuracy and $95\%$ confidence intervals are computed with 5 independent runs (5 random seeds). The backbone neural network architecture is the same for each combination of method and dataset except for the last fully-connected layer, if applicable. It is the concatenation of three modules of Convolution-BatchNorm-Relu-Maxpool. Our findings include that, for 5-way classification of OmniPrint-meta[1-5], MAML outperforms Prototypical Networks, except for OmniPrint-meta1; for 20-way classification, Prototypical Networks outperforms MAML in easier datasets and are surpassed by MAML for more difficult datasets. One counter-intuitive discovery is that the modeling difficulty estimated from learning machine performance (Figure 5 (a)) does not coincide with human judgement. One would expect that OmniPrint-meta5 should be more difficult than OmniPrint-meta4, because it involves natural backgrounds, making characters visually harder to recognize, but the learning machine results are similar.

Table 4: $N$-**way**-$K$-**shot classification results** on the five OmniPrint-meta[1-5] datasets.

| Setting | | meta1 | meta2 | meta3 | meta4 | meta5 |
|---|---|---|---|---|---|---|
| $N$=5 $K$=1 | Naive | $66.1 \pm 0.7$ | $43.9 \pm 0.2$ | $34.9 \pm 0.3$ | $20.7 \pm 0.1$ | $22.1 \pm 0.2$ |
| | Proto [58] | $\mathbf{97.6 \pm 0.2}$ | $83.4 \pm 0.7$ | $75.2 \pm 1.3$ | $62.7 \pm 0.4$ | $61.5 \pm 0.7$ |
| | MAML [15] | $95.0 \pm 0.4$ | $\mathbf{84.7 \pm 0.7}$ | $\mathbf{76.7 \pm 0.4}$ | $\mathbf{63.4 \pm 1.0}$ | $\mathbf{63.5 \pm 0.8}$ |
| $N$=5 $K$=5 | Naive | $88.7 \pm 0.3$ | $67.5 \pm 0.5$ | $52.9 \pm 0.4$ | $21.9 \pm 0.1$ | $26.2 \pm 0.3$ |
| | Proto [58] | $\mathbf{99.2 \pm 0.1}$ | $93.6 \pm 0.9$ | $88.6 \pm 1.1$ | $79.2 \pm 1.3$ | $77.1 \pm 1.5$ |
| | MAML [15] | $97.7 \pm 0.2$ | $\mathbf{93.9 \pm 0.5}$ | $\mathbf{90.4 \pm 0.7}$ | $\mathbf{83.8 \pm 0.5}$ | $\mathbf{83.8 \pm 0.4}$ |
| $N$=20 $K$=1 | Naive | $25.2 \pm 0.2$ | $14.3 \pm 0.1$ | $10.3 \pm 0.1$ | $5.2 \pm 0.1$ | $5.8 \pm 0.0$ |
| | Proto [58] | $\mathbf{92.2 \pm 0.4}$ | $\mathbf{66.0 \pm 1.8}$ | $\mathbf{52.8 \pm 0.7}$ | $35.6 \pm 0.9$ | $35.2 \pm 0.7$ |
| | MAML [15] | $83.3 \pm 0.7$ | $65.8 \pm 1.3$ | $52.7 \pm 3.2$ | $\mathbf{42.0 \pm 0.3}$ | $\mathbf{42.1 \pm 0.5}$ |
| $N$=20 $K$=5 | Naive | $40.6 \pm 0.1$ | $23.7 \pm 0.1$ | $16.0 \pm 0.1$ | $5.5 \pm 0.0$ | $6.8 \pm 0.1$ |
| | Proto [58] | $\mathbf{97.2 \pm 0.2}$ | $\mathbf{84.0 \pm 1.1}$ | $74.1 \pm 0.9$ | $56.9 \pm 0.4$ | $54.6 \pm 1.3$ |
| | MAML [15] | $93.1 \pm 0.3$ | $83.0 \pm 1.0$ | $\mathbf{75.9 \pm 1.3}$ | $\mathbf{61.4 \pm 0.4}$ | $\mathbf{63.6 \pm 0.5}$ |

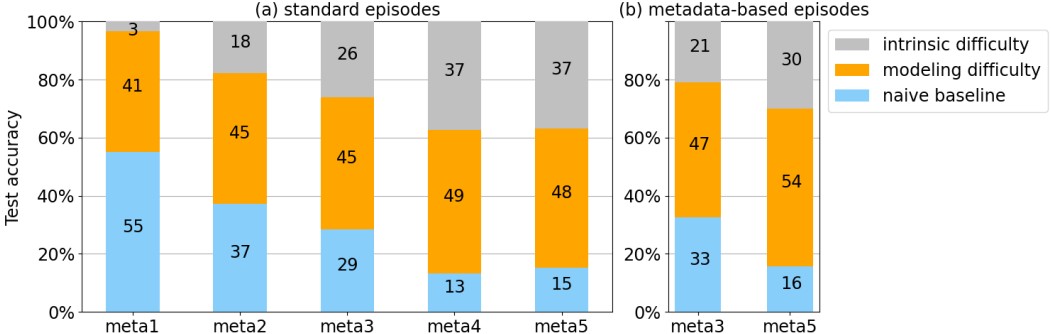

Figure 5: **Difficulty of OmniPrint-meta[1-5] (few-shot learning)**: We averaged the results of $N$-way-$K$-shot experiments of Table 4. The height of the blue bar represents the performance of the naive baseline (low-end method). The top of the orange bar is the max of the performance of Prototypical Networks and MAML (high-end methods). (a) "Standard" episodes with uniformly sampled rotation and shear. Difficulty progresses from meta1 to meta4, but is (surprisingly) similar between meta4 and meta5. (b) "Metadata" episodes: images within episode share similar rotation & shear; resulting tasks are easier than corresponding tasks using standard episodes.

## 4.2 Other meta-learning paradigms

OmniPrint provides extensively annotated metadata, recording all distortions. Thus more general paradigms of meta-learning (or life-long-learning) can be considered than the few-shot-learning setting considered in Section 4.1. Such paradigms may include concept drift or covariate shift. In the former case, distortion parameters, such as rotation or shear, could slowly vary in time; in the latter case episodes could be defined to group examples with similar values of distortion parameters.

To illustrate this idea, we generated episodes differently than in the "standard way" [58, 15, 64]. Instead of only varying the subset of classes considered from episode to episode, we also varied transformation parameters (considered nuisance parameters). This imitates the real-life situation in which data sources and/or recording conditions may vary between data subsets, at either training or test time (or both). We used the two datasets OmniPrint-meta3 and OmniPrint-meta5, described in the previous section, and generated episodes imposing that *rotation* and *shear* be more similar within episode than between episode (the exact episode generation process and experimental details are provided in Appendix I). The experimental results, summarized in Figure 5 (b), show that metadata-based episodes make the problem simpler. The results were somewhat unexpected, but, in retrospect, can be explained by the fact that meta-test tasks are easier to learn, since they are more homogeneous. This use case of OmniPrint could be developed in various directions, including defining episodes differently at meta-training and meta-test time *e.g.,* to study whether algorithms are capable of learning better from more diverse episodes, for fixed meta-test episodes.

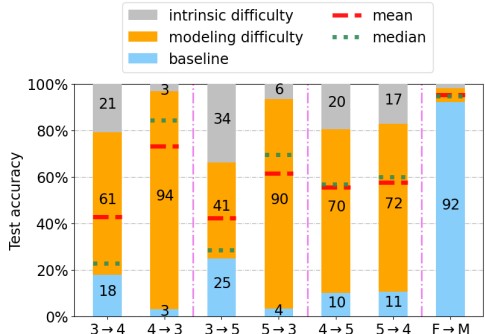

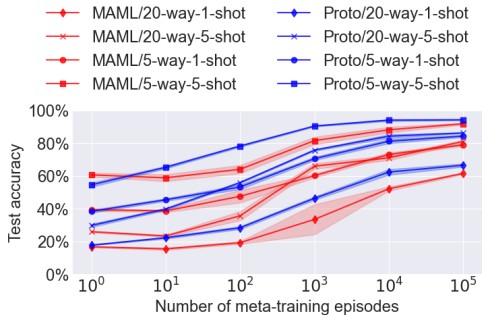

Figure 6: **Domain adaptation**. $A \rightarrow B$ means OmniPrint-metaA is source domain and OmniPrint-metaB is target domain, $A, B \in 3, 4, 5$. $F \rightarrow M$ means Fake-MNIST is source domain and MNIST is target domain. Mean and median are computed over 5 methods tried.

Figure 7: **Influence of the number of meta-training episodes** with a larger version of OmniPrint-meta3. $95\%$ confidence intervals are computed with 5 random seeds.

### 4.3 Influence of the number of meta-training episodes for few-shot learning

We generated a larger version of OmniPrint-meta3 with 200 images per class (OmniPrint-meta3 has 20 images per class), to study the influence of the number of meta-training episodes. We compared the behavior of MAML [15] and Prototypical Network [58]. The experiments (Figure 7 and Appendix J) show that the learning curves cross and Prototypical Network [58] ends with higher performance than MAML [15] when the number of meta-training episodes increases. Generally Prototypical Network performs better on this larger version of OmniPrint-meta3 than it did on the smaller version. This outlines that changes in experimental settings can reverse conclusions.

### 4.4 Domain adaptation

Since OmniPrint-meta[1-5] datasets share the same label space and only differ in styles and transforms, they lend themselves to benchmarking domain adaptation (DA) [12], one form of transfer learning [50]. We created a sample DA benchmark, called OmniPrint-metaX-31 based on OmniPrint-meta[3-5] (last 3 datasets). Inspired by Office-31 [29], a popular DA benchmark, we only used 31 randomly sampled characters (out of 1409), and limited ourselves to 3 domains, and 20 examples per class. This yields 6 possible DA tasks, for each combinations of domains. We tested each one with the 5 DeepDA unsupervised DA methods [65]: DAN [45, 63], DANN [17], DeepCoral [60], DAAN [75] and DSAN [79]. The experimental results are summarized in Figure 6. More details can be found in Appendix H. We observe that transfers $A \rightarrow B$ when $A$ is more complex than $B$ works better than the other way around, which is consistent with the DA literature [26, 16, 41]. The adaptation tasks $4 \rightarrow 5$ and $5 \rightarrow 4$ are similarly difficult, consistent with Section 4.1. We also observed that when transferring from the more difficult domain to the easier domain, the weakest baseline method (DAN [45, 63]) performs only at chance level, while other methods thrive. We also performed unsupervised DA from a dataset generated with OmniPrint (Fake-MNIST) to MNIST [39] (see Appendix H), The performance of the 5 DeepDA unsupervised DA methods range from 92% to 98% accuracy, which is very honorable (current supervised learning results on MNIST are over 99%).

### 4.5 Character image regression tasks

OmniPrint can also be used to generate datasets for regression tasks. We created an example of regression to horizontal shear and rotation. This simulates the problem of detecting variations in style that might have forensic, privacy, and/or fairness implication when characters are handwritten. OmniPrint could provide training data for bias detection or compensation models. Additionally, shear estimation/detection is one of the preprocessing steps for some OCR methods [31, 3, 6, 51, 57].

We generated two large datasets which are slightly easier than OmniPrint-meta3. Both datasets contain black-on-white characters (1409 characters with 200 images each). The first dataset has horizontal shear (horizontal shear parameter ranges from -0.8 to 0.8) but not rotation, the second dataset has rotation (rotation ranges from -60 degrees to 60 degrees) but not horizontal shear. Perspective transformations are not used. We tested two neural networks: A "small" one, concatenating

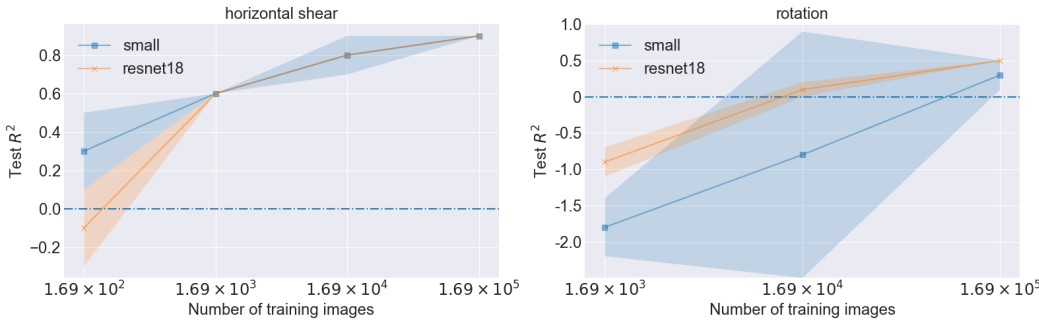

Figure 8: **Regression on text images.** $95\%$ confidence intervals are computed with 3 random seeds.

three modules of Convolution-BatchNorm-Relu-Maxpool, followed by a fully-connected layer with a scalar output (76097 trainable parameters); A "large" one, Resnet18 [25] pretrained on ImageNet [53], of which we trained only the last convolution and fully-connected layers (2360833 trainable parameters). The reported metric is the coefficient of determination $R^2$. The experiments (Figure 8 and Appendix K) show that horizontal shear is much simpler to predict than rotation.

## 5  Discussion and conclusion

We developed a new synthetic data generator leveraging existing tools, with a significant number of new features. Datasets generated with OmniPrint retain the simplicity of popular benchmarks such as MNIST or Omniglot. However, while state-of-the-art ML solutions have attained quasi-human performance on MNIST and Omniglot, OmniPrint allows researchers to tune the level of difficulty of tasks, which should foster progress.

While OmniPrint should provide a useful tool to conduct ML research *as is*, it can also be *customized* to become an effective OCR research tool. In some respects, OmniPrint goes beyond state-of-the-art software to generate realistic characters. In particular it has the unique capability of incorporating pre-rasterization transformations, allowing users to distort characters by moving anchor points in the original font *vector* representation. Still, many synthetic data generators meant to be used for OCR research put emphasis on other aspects, such are more realistic backgrounds, shadows, sensor aberrations, etc., which have not been our priority. Our modular program interface should facilitate such extensions. Another limitation of OmniPrint is that, so far, emphasis has been put on generating isolated characters, although words or sentences can also be generated. Typeset text is beyond the scope of this work.

We do not anticipate any negative societal impact. Much the contrary, OmniPrint should foster research on alphabets that are seldom studied and should allow researchers and developers to expand OCR to many many more languages. Obviously OmniPrint should be responsibly used to balance alphabets from around the world and not discriminate against any culture.

The impact of OmniPrint should go beyond fostering improvement in recognizing isolated printed characters. OmniPrint's data generative process is of the form $\mathbf{X} = f(\mathbf{Y}, \mathbf{Z})$, where $\mathbf{Y}$ is the class label (character), $\mathbf{Z}$ encompasses font, style, distortions, background, noises, etc., and $\mathbf{X}$ is the generated image. OmniPrint can be used to design tasks in which a label $\mathbf{Y}$ to be predicted is entangled with nuisance parameters $\mathbf{Z}$, resembling other real-world situations in different application domains, to push ML research. This should allow researchers to make progress in a wide variety of problems, whose generative processes are similar (image, video, sound, and text applications, medical diagnoses of genetic disease, analytical chemistry, etc.). Our first meta-learning use cases are a first step in this direction.

Further work include keeping improving OmniPrint by adding more transformations, and using it in a number of other applications, including image classification benchmarks, data augmentation, study of simulator calibration, bias detection/compensation, modular learning from decomposable/separable problems, recognition of printed characters in the wild and generation of captchas. Our first milestone is using OmniPrint for the NeurIPS2021 meta-learning challenge.

## Acknowledgments and Disclosure of Funding

We gratefully acknowledge many helpful discussions about the project design with our mentors Anne Auger, Feng Han and Romain Egele. We also received useful input from many members of the TAU team of the LISN laboratory, and the MetaDL technical crew: Adrian El Baz, Bin Feng, Jennifer (Yuxuan) He, Jan N. van Rijn, Sebastien Treguer, Ihsan Ullah, Joaquin Vanschoren, Phan Anh Vu, Zhengying Liu, Jun Wan, and Benjia Zhou. OmniPrint is based on the open source software TextRecognitionDataGenerator [2]. We would like to warmly thank all the contributors of this software, especially Edouard Belval. We would like to thank Adrien Pavao for helping providing computing resources. We would also like to thank the reviewers for their constructive suggestions. This work was supported by ChaLearn and the ANR (Agence Nationale de la Recherche, National Agency for Research) under AI chair of excellence HUMANIA, grant number ANR-19-CHIA-0022.

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
