# OpenReview forum: "OmniPrint: A Configurable Printed Character Synthesizer"
_NeurIPS.cc/2021/Track/Datasets_and_Benchmarks/Round1 — NeurIPS 2021 Datasets and Benchmarks Track (Round 1)_

### Official Review · Reviewer_Hhus · 2021-06-28
**An interesting character image synthesis tool for the research in few-shot learning and OCR**

**Rating:** 7
**Confidence:** 3
**Clarity:** The paper is well written.

**Strengths:**

1. Many pre-rasterization transformations are implemented in OmniPrint for a large variety of character styles, which makes the synthetic data more realistic and difficult.

2. The experiments on few-shot learning tasks demonstrated that OmniPrint can generate datasets with a good control of difficulty levels, and are more difficult to tackle than the previous Omnilglot dataset.

3. I think the proposed character image generator with full control of various transformations can be useful for research on few-shot learning tasks and other OCR problems.


**Weaknesses:**

1. Since it is claimed that the proposed synthetic data generator can be used for many use cases, it would be better to include experiments on several use cases. Currently, only one use case for meta-learning is considered. I’m especially interested in how the synthesized dataset benchmarks the SOTA methods on the font style recognition (i.e., regression problems).

2. For experiments on few-shot learning tasks, how does the training data size (i.e., the number of meta-train tasks) impact the results? Since the proposed generator can synthesize as many images as possible, it would be great to show if providing more data to meta-learning methods would largely increase the performance.

___________________
post-rebuttal update:
Thank the authors for providing additional experiments and use cases. The current version looks better so I raise my score to 7 and recommend the acceptance.

**Additional Feedback:**

N/A

**Correctness:**

The dataset is constructed in a sound way. The evaluation methods and experiment design overall look good. The only concern is that only two meta-learning methods have been tested, so I’m not sure if the same conclusions hold for other meta-learning methods.

**Documentation:**

There is sufficient detail on documentation.


**Ethics:**

The ethical concerns have been discussed thoroughly in the paper.

**Relation To Prior Work:**

It clearly discussed how this work differs from previous contributions.

**Summary And Contributions:**

This work proposes a new character image generator called OmniPrint, that generates a wide variety of characters from more than 20 languages, together with different fonts, styles, textured backgrounds and noises, and various pre-rasterization transformations. By using OmniPrint, five few-shot learning tasks (named OmniPrint-meta[1-5]) with increasing difficulty are generated to benchmark meta-learning methods.

---

### Official Review · Reviewer_71Yz · 2021-07-03
**A New Synthetic Printed Character Generator Leveraging Existing Tools With a Significant Number of New Features**

**Rating:** 6
**Confidence:** 3

**Strengths:**

- The introduced printed character generator is interesting. The authors demonstrate their effort in creating a comprehensive data synthesizer with various useful features, covering diverse languages, fonts, transformations, styles, *etc*.

- The documentation and information of the data synthesizer are provided in detail.

- The printed character synthesizer may encourage future work on various machine learning problems, such as image classification, data augmentation, meta-learning, transfer learning, Optical Character Recognition, *etc*.

- The authors present an example use case of meta-learning to verify the usefulness of the proposed OmniPrint.



**Weaknesses:**

- The proposed OmniPrint data synthesizer is built based on the open-source software "Text Recognition Data Generator". Nevertheless, a direct comparison of OmniPrint with this software is missing. It would be better to present sufficient comparisons (experimentally for the best) to demonstrate the superiority of OmniPrint over "Text Recognition Data Generator".

- The authors claim many use cases and applications of OmniPrint, while only one of these use cases (*i.e.*, meta-learning) has been presented and supported.

- Although necessary details of the proposed data synthesizer have been provided, the writing of some parts of this paper could be further improved and better structured: 1) It would be better to reorganize or rewrite the paragraph starting from Line 25 in Section 1. The core idea of this paragraph is unclear, and the structure looks not very clean. 2) The paragraph starting from Line 136 is too long. This might make readers hard to follow the steps and details in this paragraph.

**Additional Feedback:**

- Line 18 Typo: "Optical Characted Recognition" -> "Optical Character Recognition".

- Line 41 Typo: "Y and Z and" -> "Y and Z are".

**Clarity:**

The paper is generally well-written and explained in detail. Some writing issues can be found in Weaknesses.

**Correctness:**

Most of the claims in the submission are correct. Please refer to Weaknesses for some of my concerns.

**Documentation:**

The authors have tried their best to provide details of their proposed data synthesizer. The documentation has been given in detail.

**Ethics:**

From my perspective, there are few or no ethical concerns that warrant further discussion or review.

**Relation To Prior Work:**

The paper discussed how this work differs from previous contributions. It would be better to make a direct comparison with the open-source software this work is based on.

**Summary And Contributions:**

This paper introduces OmniPrint, a new synthetic data generator of isolated printed characters, which is built on existing tools with a significant number of new features. OmniPrint offers the capability of generating a wide variety of printed characters from various languages, fonts, and styles, with customized distortions. The authors list a wide range of use cases and applications of OmniPrint. They show an example use case of meta-learning to verify the usefulness of OmniPrint toward machine learning research.

---

### Official Review · Reviewer_qx9T · 2021-07-04
**Good contribution to meta-learning, but not necessarily to OCR**

**Rating:** 7
**Confidence:** 2

**Strengths:**

- **[Major] More diverse and challenging than prior datasets (MNIST and Omniglot).** This dataset provides an extensible framework for generating more challenging characters for ML tasks. Most importantly, the synthesized characters are much more diverse than previous work: OmniPrint supports multiple languages other than Latin characters and Chinese, multiple fonts, and arbitrary pre-rasterization transformations. I think this dataset could be very useful for basic meta-learning research and potentially OCR.
- **[Minor] Extensible.** The authors have done a good job of making the framework extensible to new character types - conceivably OmniPrint can be easily extended to other fonts and languages, and the variety of transformations provided is promising.

**Weaknesses:**

- **[Major] Real-world problem?** One desirable property of a dataset for research is that it reflects a real-world problem. OmniPrint is potentially useful for real-world OCR tasks, but it seems more geared towards making meta-learning benchmarks more challenging. I see the value in this contribution, but I am not sure the increased challenge is well-aligned with a real-world application. The authors list only a few **concrete** applications (recognition of printed characters in the wild, creation of catpchas) outside **abstract** learning problems like transfer learning and modular learning (L5-7).

    I can see that recognition in the wild is a potential application of OmniPrint, but this contribution isn't the primary goal of the dataset. As the authors acknowledge, only single characters are supported, and while textured backgrounds are included, more realistic backgrounds & other aberrations are not the priority (L297-300). As the authors suggest (L293-294), OmniPrint could potentially be further improved (with extra focus on shadow, realistic backgrounds, multi-character sequences, etc.) to make it a competitive in-the-wild generator, but as it stands I don't think there is a significant contribution to "in the wild" characters.

    There isn't any further discussion of the captcha generation application, so it seems that the primary contribution is to meta-learning - not OCR.

EDIT (07/16/2021): with the reframing towards meta-learning and the new examples, this concern has been addressed.

**Additional Feedback:**

Though I think the contributions should be framed differently, I think this dataset is a valuable contribution to ML research and could be useful to many developers.

**Clarity:**

The paper is well-written and clear throughout. I especially appreciate the Figures - Figure 5 is an intuitive explanation of the increase in difficulty; Figure 2 clearly explains the character generation process and the various phases of the synthesizer.

I see one typo in L27 - "though" instead of "thought".

**Correctness:**

As far as I can tell, the dataset is soundly constructed. The examples are well taken, and on first glance the code repository is well-documented and usable. I'm not sure how useful the background textures are - especially since adding them didn't seem to make much of a difference to performance in the case study (L277-278).

**Documentation:**

The documentation seems straightforward and understandable. The synthesization process seems reproducible as well.

**Ethics:**

I don't see any problematic uses of this dataset or issues with the collection process, since the data is synthetic. The authors may consider consulting native speakers of all the languages included to make sure they are accurately represented, if they haven't already.

**Relation To Prior Work:**

**Pro:** Table 1 clearly contrasts OmniPrint and Omniglot - I think this is a nice, concise summary of OmniPrint's advantages over previous printed character datasets.

**Con:** As mentioned in the Weaknesses section, the primary contribution of this paper seems to be to meta-learning, not OCR. For this reason I hoped the authors would spend more time discussing related work in meta-learning, rather than its relation to previous text synthesizers for OCR.

EDIT (07/16/2021): this con has *mostly* been addressed. The authors did some reframing towards meta-learning work but the related work section is still mostly OCR.

**Summary And Contributions:**

This paper proposes a object classification benchmark dataset for modeling the speed-accuracy trade-off in human cognition. The authors use the benchmark to compare human cognition to neural network behavior, providing both a model for the human speed-accuracy trade-off and recommendations for computer vision research.

By way of framing this review: I am familiar with computer vision research in general, but I have very little knowledge of human cognition research of this kind.

---

### Decision · Program_Chairs · 2021-07-26

**Decision:**

Accept

**Comment:**

Inspiring from famous machine learning datasets such as MNIST, SVHN and omniglot, this paper proposes a synthetic data generator of characters to generate a wide variety of printed characters from various languages, fonts and styles. The dataset mainly focuses on meta learning.

This paper proposes a more diverse and challenging than prior datasets as a challenge for meta-learning besides traditional ML datasets such as MNIST and omniglot. The dataset is well-thought out and designed.

All the reviewers agreed that the paper is worth publishing with minor considerations. The authors did a good job addressing most of the concerns. Thus I recommend this paper for acceptance.